# Algorithms for the Motion of Randomly Positioned Hexagonal and Square Microparts on a “Smart Platform” with Electrostatic Forces and a New Method for Their Simultaneous Centralization and Alignment

**DOI:** 10.3390/mi10120874

**Published:** 2019-12-12

**Authors:** Georgia Kritikou, Nikos Aspragathos, Vassilis Moulianitis

**Affiliations:** 1Robotics Group, Department of Mechanical Engineering and Aeronautics, University of Patras, 265 04 Rio, Greece; asprag@mech.upatras.gr; 2Department of Product and Systems Design Engineering, University of the Aegean, 84100 Syros, Greece; moulianitis@syros.aegean.gr

**Keywords:** electrostatic, smart platform, microparts, SEA, MEA, collision avoidance, neighboring microparts, parallel, centralization, aligning

## Abstract

In this paper, an approach is proposed for the simultaneous manipulation of multiple hexagonal and square plastic–glass type microparts that are positioned randomly on a smart platform (SP) using electrostatic forces applied by the suitable activation of circular conductive electrodes. First, the statics analysis of a micropart on the SP is presented in detail and the forces and torques that are applied to and around the center of mass (COM) respectively due to the activation of a SP electrode are determined. The “single electrode activation” (SEA) and the “multiple electrodes activations” (MEA) algorithms are introduced to determine the feasible SP electrodes activations for the microparts manipulation considering their initial configuration. An algorithm for the simultaneous handling of multiple microparts is studied considering the collision avoidance with neighboring microparts. An approach is presented for the simultaneous centralization and alignment of the microparts preparing them for their batch parallel motion on the SP. The developed algorithms are applied to a simulated platform and the results are presented and discussed.

## 1. Introduction

The manipulation of the microcomponents building micro-electromechanical systems (MEMS) is an area, which attracts the interest of many researchers globally. In the MEMS industry, these procedures are implemented serially with microgrippers [1,2,3], microtweezers [4,5], and microrobots [6,7], techniques which are not suitable for low cost parallelization of the production. According to many statistical analyses that have been presented in the relevant bibliography, these methods considerably increase the cycle-time production in the microfactories [8,9,10]. The contactless micromanipulation with forces that overcome the weight of the microcomponents in the micro world is an upcoming solution promising to make the microparts handling more autonomous. 

Boheringer et al. [11] proposed a sensorless technique for the contactless micromanipulation fields of nonsymmetric microparts with programmable force on an array with mechanical micro-actuators. The mathematical formula of three particular force fields for the serial centralization, alignment, and transfer of two microcomponents in order to be assembled was presented [12]. These force fields were applied to convex and nonconvex micropart on a mechanical micro-actuator system [13]. A platform with mechanical actuators has also been used for the handling of a convex microcomponent with programmable force fields trapping the micropart in regions of the field’s local minimum [14]. The formula of the programmable force fields is specified considering the geometry of the microparts, their start position, and their start orientation [11,14]. But, the status of the force fields that are applied in the whole handling region has to change for each micromanipulation step in order to be customized in the shape and the configuration of a single micropart. This constraint complicates the parallel manipulation of multiple different-shaped, randomly-oriented microcomponents. 

Thus, more efficient contactless microhandling methods for the simultaneous manipulation of the components is needed. The parallel trapping of conductive [15] and polymer [16] mili objects on vibrating plates with electrostatic forces has been proposed. Sun et al. [17] simulated the 3D motion of paramagnetic spheres in regions with magnetic fields while in [18] the levitation and micromanipulation of solid and liquid micromaterials that are enclosed in paramagnetic droplets and handled in magnetic fields, was presented. 

Methods for the microparts contactless manipulation in special designed platforms have also been proposed. Techniques have been developed for the planar motion of the micropart due to the micro air-injectors that are placed underneath the manipulation surface of the platform [19,20]. On this device, the micropart is pushed by the air and it is driven to a new equilibrium position with low accuracy due to the high magnitude of the applied force. Dkhil et al. [21] introduced a platform for the quick traverse of the microparts on an air/liquid surface with magnetic fields. The discrete motion of a magnetic microrobot on a platform with embedded microcoils was validated in [22]. However, when a second microrobot starts its parallel motion on the platform, the intensity of the magnetic forces that are applied to the robots gets increased, trapping them on the activated coils. 

The microprocessors [23], the microgenerators [24] etc., that compose the MEMS are integrated circuits in plastic/insulating materials. Kritikou et al. [25] proposed a “smart platform” (SP) with embedded conductive electrodes and visual sensors for the handling of multiple plastic/plexiglass microcomponents with electrostatic forces. The motion of the centralized and aligned microparts between two equilibrium positions was implemented due to the activation of specified SP electrodes [26]. But, the microparts that are going to be manipulated on the SP are transmitted to it with transfer lines [8]. As a result, after their initial positioning on the SP, the micro objects are not centralized and aligned (randomly positioned). A path computation method for the parallel motion of multiple microparts on the simulated SP for centralized and aligned microparts has been proposed [25], neglecting how the microparts left their random start configuration. According to the authors’ knowledge, there are no works including activation methods for the motion of the randomly positioned microparts between two equilibrium positions and for their parallel centralization and aligning on the SP. But, in order to be considered the SP as an autonomous–flexible device, the handling of the randomly positioned microcomponent on it is needed. 

In this work, the demanding issue of the micromanipulation of non-centralized and non-aligned (randomly positioned) microparts on the SP is studied. Firstly, the static analysis of a randomly positioned micropart on the SP is performed and the forces and torques that are applied to its center of mass (COM) due to a single activation of a SP electrode are determined. Two new algorithms are introduced: the single electrode activation (SEA) and the multiple electrodes activation (MEA). The algorithms compute all the activation combinations that can be applied to the SP electrodes and specify the motion that can implement the micropart due to them. The translation of a randomly positioned micropart between two equilibrium positions applying the activations that result from SEA and MEA algorithms is simulated. A new method for the simultaneous centralization and alignment of multiple static randomly positioned microparts avoiding the collision between them is proposed. 

The remaining of the paper is organized as follows: In the following section, the contribution of the paper is presented, in Section 3 a brief description of the smart platform and the layout of the electrodes are presented and the statics analysis of the microparts due to a single SP electrode activation is studied. In Section 4, the single electrode activation (SEA) and the multiple electrodes activation (MEA) algorithms are presented in details. A method for the automated centralization and aligning of the microparts is presented in Section 5. The results of the algorithms and the methods that the authors introduced in this paper are discussed in Section 6, where their future work is also included. In Table 1 is shown the description of the symbols that are going to be used in the rest of the paper. 

## 2. The Contribution 

The SP is a device for the manipulation of the microparts with electrostatic forces. At the beginning of their transfer on the SP, the microparts reach random configurations. However, there are no references proposing methods for the microparts’ motion and their automated centralization and alignment on the SP. But, in order to be considered as an efficient and autonomous device, activation methods for the displacement of the randomly positioned microparts are needed. Moreover, the path planning for the simultaneous motion of multiple microparts on the SP can be applied only to centralized and aligned components. Thus, the main contribution of this work is summarized in the following two points:The computation of the activation for the motion of the randomly positioned microparts on the SP.A method for the simultaneous centralization and alignment of multiple microparts on the SP avoiding the collision between them.

## 3. The Smart Platform and the Microparts 

The “smart platform” (SP) is a Printed Circuit Board (PCB) with circular grounded conductive electrodes and two dielectric layers on top that are positioned according to the cross-section shown in Figure 1 [23]. On the top of this microelectromechanical system (MEMS), the insulating glass-type material microparts with circular conductive underneath, can be manipulated due to the electrostatic forces that are presented between the adjacent charged SP and micropart electrodes. The position of the microparts are detected with the aid of the vision system. Then, as it is shown in Figure 1, the SP electrodes get charged, when a constant positive potential is applied to them. As a result, the micropart’s electrodes that are positioned close to the charged area, get charged negatively due to the polarization of the dielectric layers. Finally, the micropart electrode is attracted by the corresponding charged SP electrode, and in the case that the electrostatic forces between them are strong enough, the micropart leaves its current equilibrium position and moves to reach a new one. The electrostatic force between a charged SP electrode and a corresponding micropart electrode were studied and semi-empirical models for the force function with respect to their centers planar distance (d) is presented in [25,26,27]. 

As is shown in Figure 2, the SP has a square form with a side length equal to $L_{P},$ where the Global Coordinate System (GCS) coincides with the SP’s center of symmetry. The conductive electrodes create a grid where, $\left(x_{n},\;y_{n}\right)$ the coordinates of the $n^{th}$ SP electrode, which is defined according to the columns and lines counting method for a two dimensional array: n=c+(l−1)N, ∀c,l∈[1 N], where c is the column and the *l* line, respectively [28]. The electrodes are placed on the SP following a layout so that their centers’ horizontal/vertical Euclidean distance [29] $d_{e}$ is equal to $2\cdot r+d_{1}r,$ where r is the radius of the electrodes. The $d_{1}$ is a coefficient to adjust the distance between the centers of electrodes so that the amplitude of the electrostatic forces is enough for moving the microparts, while the undesirable arc phenomenon to be avoided [30]. The value of distance $d_{e}$ is determined so that $d_{1}\;\in \left[0.5\;0.896\right]$. The value of r is selected considering the dimensions of the micropart in order to limit the drag forces that are presented between the liquid dielectric layer and the micropart. The SP is used for the contactless micromanipulation exclusively with electrostatic forces without mechanical vibrations as in [15,16,31]. Thus, the limitation of the magnitude of the adhesive forces, which are more intense when the size of the microparts gets increased [32], is achieved when the length $L_{m}=6\cdot r+3d_{1}r$ is less than 500 μm. So, t the microparts and the SP electrode radius have to be ranged between: 1 μm ≤r≤ 60 μm.

The five circular conductive electrodes are placed underneath the hexagonal and square plastic/plexiglass microparts formatting the square corners layout, which is proposed in [26]. The planar forces that are presented between the micropart and the SP activated electrodes, resulting in its motion are the outgrowth of the successive activation of the micropart’s adjacent SP electrodes. In Figure 2, the activation methods for the horizontal, vertical, and diagonal motion of the centralized and aligned static microparts between two equilibrium positions are illustrated [25,26]. 

The total attractive electrostatic force between the electrodes is the sum of the $F^{i}{}_{v}$ and $F^{i}{}_{v,\;ver}$ vectors shown in Figure 1, where the $F^{i}{}_{v}$ represents the planar electrostatic force and the $F^{i}{}_{v,\;ver}$ the vertical, respectively. The $F^{i}{}_{v}$ and $F^{i}{}_{v,\;ver}$ with respect to the charged SP electrode and micropart’s electrode centers horizontal distance (d) (interaction in the red ellipse in Figure 1), is studied using FEM analysis for three different radiuses of the electrodes. The radius of the SP and micropart electrodes is equal to 15, 25, and 60 μm,
$d_{1}=0.7$ so that $d_{e\;}\left(15\mu m\right)=40\;\mu m,$
$d_{e\;}\left(25\;\mu m\right)=67.5\;\mu m$ and $d_{e\;}\left(60\;\mu m\right)=160\;\mu m$ and the voltage that is applied to each is equal to 50, 50, and 75 V, respectively [25]. The solid dielectric $\mathrm{Si}O_{2}$ with $\varepsilon_{rs}=6$ and the oil lubricant with dielectric properties due to the $Al_{2}O_{3}$ nanoparticles oil additives [33] with dielectric constant equal to $\varepsilon_{rl}=8$ [34] and static force coefficient $\mu_{s}=0.3$ [35] are used. In Figure 3, the determinations of the planar electrostatic force between the charged SP electrode and the micropart electrode with respect to d are shown. 

The micropart leaves its static condition when:(1)$$\overset{\kappa }{\underset{\kappa^{\prime }=1}{\sum }}F^{i}{}_{v}\left(d\left(n^{\kappa^{\prime }},\;i_{v}\right)\right)>F_{s}=\mu_{s}\cdot \left(\overset{\kappa }{\underset{\kappa^{\prime }=1}{\sum }}F^{i}{}_{v,ver}\left(d\left(n^{\kappa^{\prime }},\;i_{v}\right)\right)+W\right),$$
where κ is the total number of the simultaneously activated SP electrodes, $n^{\kappa^{\prime }}\;\mathrm{is}$ the activated SP electrode that interacts with the $i_{v}$ electrode of the $m_{i}$ micropart ∀v∈[1 5],
$F_{s}$ is the static friction force between the micropart and the dielectric lubricant that is equal to $\mu_{s}\cdot \left(\sum_{\kappa^{\prime }\;=\;1}^{\kappa }F^{i}{}_{v,ver}\left(d\left(n^{\kappa \prime },\;i_{v}\right)\right)+W\right)$ where W is the weight of the micropart. Considering that the weight W of the micropart is a negligible value the contribution of the $\mu_{s}\cdot \sum_{\kappa^{\prime }\;=\;1}^{\kappa }F^{i}{}_{v,ver}\left(d\left(n^{\kappa^{\prime }},\;i_{v}\right)\right)$ is compared to $F^{i}{}_{v}.$ In Figure 4, the vertical electrostatic force determinations multiplied by $\mu_{s}$ with respect to d is shown.

Considering the graphs in Figure 3 and Figure 4, the mode of the planar electrostatic force with respect to the electrodes distance differs significantly from the corresponding function of $\mu_{s}\cdot \sum_{\kappa^{\prime }\;=\;1}^{\kappa }F^{i}{}_{v,ver}\left(d\left(n^{\kappa^{\prime }},\;i_{v}\right)\right)$ vs. d. The overlapping between the microparts and SP electrodes starts when d<2r, where as it is shown in Figure 3 and Figure 4, the planar electrostatic force is significantly greater than the $\mu_{s}\cdot F^{i}{}_{v,ver}\left(d\left(n,\;i_{v}\right)\right)$ for d>0.1r. Specifically, when the overlapping between the electrodes is bigger than $\frac{\left(2r-0.1r\right)}{2r}\cdot 100\%=\frac{120-6\mu m}{120\mu m}≅\frac{\left(50-2.5\right)\mu m}{50\mu m}≅\frac{\left(30-1.5\right)\mu m}{30\mu m}>95\%,$ the $F^{i}{}_{v}$ and $\mu_{s}\cdot \sum_{\kappa^{\prime }\;=\;1}^{\kappa }F^{i}{}_{v,ver}\left(d\left(n^{\kappa^{\prime }},\;i_{v}\right)\right)$ decreases and increases significantly, respectively. But, the intensity of the vertical electrostatic contributes in the increase of the adhesive forces that trap the micropart on the SP [36]. However, the micropart and the activated SP electrodes are a couple of positively and negatively charged plates with two dielectrics which temp to be self-assembled and centralized since their overlapping is bigger than 0% [37]. So, the micropart electrode can be centralized on an activated SP electrode when $0\%<\frac{\left(2r-d\right)}{2r}\cdot 100\%<95\%.$ In the case that the overlapping between the electrodes is bigger than 95%, the micropart electrode is considered as centralized on the activated SP electrode. However, the micropart can be displaced due to a single SP electrode activation leaving its static condition when $0.1r<d\leq d_{e}.$ The $F^{i}{}_{v}\left(d\left(n,\;i_{v}\right)\right)$ and the $F^{i}{}_{v,ver}\left(d\left(n,\;i_{v}\right)\right)$ are described by a 6th grade and 3rd grade polynomial fitting function, respectively, which are used for the computations in the following Sections. 

A demanding issue is the activation methods that are applied to the SP electrodes for the motion of the microparts between two equilibrium positions when they are not centralized (the micropart electrode underneath the COM of the micropart does not coincide with a SP electrode) and non-aligned with the GCS (randomly positioned microparts). The position of the microparts on the SP is detected by the vision system and its orientation is described with the template matching method [38]. In Section 3, the static analysis of a randomly positioned micropart is described in order to be used in the activation methods computation which will be discussed in the following sections. 

In Figure 5, a part of the SP region and the square corners layout underneath the $m_{i}$ randomly positioned micropart is illustrated. Each electrode’s configuration $q^{i}=\left(r^{i},\;\theta^{i}\right)$ is computed considering the vector $r^{i}=\left(x^{i},\;y^{i}\right)$ and the angle $\theta^{i}$ that define the COM’s position and the micropart’s orientation with respect to the GCS, respectively. The micropart can reach any position on the SP apart from these at the edges of the SP, since it is driven out of the SP. Thus, the coordinates of the $r^{i}$ have to satisfy the constraints: (2)$$-\frac{L_{P}}{2}+d_{e}\leq x^{i}\leq \;\frac{L_{P}}{2}-d_{e}\;\mathrm{and}\;-\frac{L_{P}}{2}+d_{e}\leq y^{i}\leq \;\frac{L_{P}}{2}-d_{e}.$$

The five electrodes are positioned on the diagonals of their square corners layout so that when the COM of the micropart coincides with the center of a SP electrode and $\theta^{i}=0$ all the micropart electrodes to coincide with a SP electrode, respectively. The position of the center of the $v^{th}$ micropart electrodeis given by: (3)$$\begin{array}{c} x_{i_{1}}=x^{i},y_{i_{1}}=y^{i},\;v=1,x_{i_{v}}=x^{i}+\sqrt{2}d_{e}\sin\left(\theta_{i_{v}}\right),\;\\ y_{i_{v}}=y^{i}+\sqrt{2}d_{e}\cos\left(\theta_{i_{v}}\right)\forall v\;\in \left[2\;5\right], \end{array}$$
where the value of each $\theta_{i_{v}}$ is included in Table 2.

The planar forces and torques that are applied to a micropart with the square corners layout, due to the single interaction between an activated SP electrode and an adjacent micropart electrode, is studied in this section. A horizontal single interaction is shown in Figure 6a, where the local coordinate system $LCS^{i}$ of the micropart is represented with the red dashed vectors, the purple dashed line corresponds to one of the two diagonals of the square corners layout, and the purple star represents the center of the $i_{v}$ micropart electrode. The pink star is the center of an activated SP electrode (n), which has sufficient distance from the micropart electrode so to interact with it, with an electrostatic force $F^{i}{}_{v}\;\left(d\left(n,\;i_{v}\right)\right)$ (blue vector–symbol $F^{i}{}_{v}$ in Figure 6b,c. The statics analysis of the micropart and the contribution of the $F^{i}{}_{v}$ in its translation is shown in Figure 6b, where the $F^{i}{}_{v}$ is analyzed to its components that are determined by: (4)$$F^{ix}{}_{v}\left(d\left(n,\;i_{v}\right)\right)=\left|F^{i}{}_{v}\left(d\left(n,\;i_{v}\right)\right)\right|\cdot \cos\left(\theta^{elec}-\theta^{i}\right),$$
(5)$$F^{iy}{}_{v}\left(d\left(n,\;i_{v}\right)\right)=\left|F^{i}{}_{v}\left(d\left(n,\;i_{v}\right)\right)\right|\cdot \sin\left(\theta^{elec}-\theta^{i}\right).\;$$

As it is shown in Figure 6c, the $F^{i}{}_{v}$ vector is analyzed in two components where the $F^{i//}{}_{v}$ is parallel and the $F^{i┴}{}_{v}$ is perpendicular to the diagonal. The $F^{i┴}{}_{v}$ contributes in the rotation of the micropart and the torque is computed by:(6)$$\vec{\tau^{i}{}_{v}}=\vec{\alpha }\times \vec{F^{i}{}_{v}}\;=\left(\sqrt{2}d_{e}\right)\cdot \left|F^{i}{}_{v}\left(d\left(n,\;i_{v}\right)\right)\right|\;\cdot \;\cos(45^{ο}\;+\left(\theta^{elec}-\theta^{i}\right)),\;\;\;\;$$
where $\vec{a}$ a vector with magnitude equal to the distance between the COM of the micropart and the center of the micropart electrode parallel to the diagonal of the micropart. The value $45^{ο}+\left(\theta^{elec}-\theta^{i}\right)$ for the analysis of the $F^{i┴}{}_{v}$ component is computed considering the straight lines $\delta_{1}$ and $\delta_{2},$ which are parallel to the $r^{i}$ vector crossing the $LCS^{i}$ and the diagonal. The straight line $\delta_{3}$ is parallel to the $F^{i┴}{}_{v}$ and its cross-section with $\delta_{1}$ and $\delta_{2}$ results in the DCB triangle. The diagonal is also the bisectrix of the $LCS^{i},$ thus $\hat{\mathrm{CBD}\;}=\;45^{ο}$ and since the DCB is an orthogonal triangle $\left(\hat{\mathrm{BDC}\;}=90^{ο}\right),$ the angle $\hat{\mathrm{DCB}\;}$ is equal to $45^{ο}$ (the sum of the angles of a triangle theorem). The angles $\hat{\mathrm{EDC}\;}$ and $\hat{\mathrm{DCB}\;}$ are equal as the inside alternate angles, thus $\hat{\mathrm{EDC}\;}=45^{ο}$ and as it is shown in Figure 6c, the remaining is the $\theta^{elec}-\theta^{i}$ angle between $F^{i}{}_{v}$ and $r^{i}.$

## 4. Activation Algorithms for the Manipulation of Randomly Positioned Microparts 

In this section, the activations of the SP electrodes and the forces and torques that are applied to the COM of a static adjacent randomly positioned micropart due to them, are determined. A micropart leaves its equilibrium position in order to reach a new one on the SP implementing: translational motion when $\left|\sum F^{i}\right|>F_{s}$ and $\left|\sum \tau^{i}\right|<T_{s},$rotational motion around a fixed axis that passes through the COM of the micropart when $\left|\sum F^{i}\right|<F_{s}$ and $\left|\sum \tau^{i}\right|>T_{s},$ and a planar motion $\left|\sum F^{i}\right|<F_{s}$ and $\left|\sum \tau^{i}\right|<T_{s}$ when where $\sum F^{i}$ and $\sum \tau^{i}$ is the sum of the planar force and torque that is presented to its COM and $F_{s}$ and $T_{s}=\vec{F_{s}}\times \vec{a},$ the static friction force and torque, respectively. 

In this section, all the feasible SP electrodes activation methods that applied to a static randomly positioned micropart are studied with the aid of two new algorithms the single electrode activation and multiple electrodes.

### 4.1. Motion of Microparts due to a Single SP Electrode Activation

The “single electrode activation” (SEA) algorithm is introduced in this section and the corresponding flow chart is illustrated in Figure 7. The input of the algorithm is the configuration $q^{i},$ the position of the micropart electrodes given by Equation (3) and Table 2 and the group $D_{1n}.$ As it is shown in Figure 8, the group $D_{1n}$ includes the electrodes of a disc area $D_{1}$ around the micropart’s COM where, $\left(x_{n},\;y_{n}\right)\in D_{1}\;\to \left[n,\;\sigma_{1}\right]\in D_{1n},$
$\sigma_{1}\in \left[1\;\sigma \right],$ while σ is the total number of electrodes in $D_{1n}.$ So, for the $\sigma_{1}^{th}$ electrode of the $D_{1n}$ and for each of the micropart’s electrodes $i_{v}$ if $0.1<d\left(n,i_{v}\;\right)\leq d_{e}$ the $F^{i}{}_{v}\left(d\left(n,i_{v}\;\right)\right)$ and its components are computed using Equation (4), (5), and (6). The components are added to $\sum F^{i},\;\sum F^{i}{}_{v,ver},$ and $\sum \tau^{i},$ since as it is illustrated in Figure 8, a SP electrode activation can influence more than one micropart electrode. Then, it is checked out if the total electrostatic force and torque that is applied to the COM of the micropart is greater than the static force and torque that is presented between the SP and the micropart. The list given by Equation (7) is included in the group $S_{m_{i}}$ of the SEA algorithm: (7)$$\left[\sum F^{ix},\;\sum F^{iy},\;\sum \tau^{i},n,\sigma_{1}\;,u\right]\;\in S_{m_{i}}\left(\sigma_{1}\right),\forall n\in D_{1n},u\in \left\{0,1,2,3\right\}.\;$$

The list includes a SP electrode of the $D_{1n}$ and the forces and torques that are applied to the static randomly positioned micropart $m_{i}$ due to its activation. The indicator u is specified considering the constraints that are included in Table 3.

The output of the algorithm in Figure 7 is the group $S_{m_{i}}$ where its lists are filtered with respect to the motion that is needed to implement the micropart. 

### 4.2. Motion of the Microparts due to the Combination of Multiple SP Electrodes Activations

In this section, an algorithm for the determination of the combination for the simultaneous activation of multiple SP electrodes called the “multiple electrodes activations” (MEA) algorithm, is introduced. The MEA algorithm is used in the case that the activations given by the SEA in the $D_{1}$ disc area are not sufficient to move the micropart as it is requested (e.g., to move it toward a specific direction). The input of the MEA is the group of the σ lists of the $S_{m_{i}}$ group that have been already computed by the SEA algorithm for the SP electrodes in the $D_{1}$ disc area around the COM of the micropart (Figure 8). The combinations of the electrodes $n\;\in \;S_{m_{i}}$ taken κ at a time where $\kappa \in \left[2\;\kappa_{1}\right],$
$3\leq \kappa_{1}\leq \sigma ,$ are determined. Specifically, as it is shown in the flow chart in Figure 9, the $\kappa_{1}$ indicator is specified and $\forall \kappa \in \left[2\;\kappa_{1}\right],$ the $\omega_{2}=\left(\begin{array}{c} \sigma \\ \kappa \end{array}\right)$ combinations of the σ lists, are specified and $\forall \;\omega_{2}^{\prime }\in \left[1\;\omega_{2}\right]$ the algorithm computes the corresponding list:(8)$$\begin{array}{c} \left[\begin{array}{c} \overset{\kappa }{\underset{\kappa^{\prime }=1}{\sum }}\left(\sum F^{ix}n^{\kappa^{\prime }}\left(\omega_{2}^{\prime }\right)\right),\;\overset{\kappa }{\underset{\kappa^{\prime }=1}{\sum }}\left(\sum F^{iy}n^{\kappa^{\prime }}\left(\omega_{2}\prime \right)\right),\;\overset{\kappa }{\underset{\kappa^{\prime }=1}{\sum }}\left(\sum \tau^{i}n^{\kappa^{\prime }}\left(\omega_{2}\prime \right)\right),\\ \left[n^{1}\left(\omega_{2}\prime \right)\ldots \;n^{\kappa^{\prime }}\left(\omega_{2}\right)\ldots n^{\kappa }\left(\omega_{2}\right)\right],\;u \end{array}\right],\\ \in M_{m_{i}},\forall n^{\kappa^{\prime }}\left(\omega_{2}^{\prime }\right)\in D_{1n},u\in \left\{0,1,2,3\right\}, \end{array}$$
where $\sum_{\kappa^{\prime }=\;1}^{\kappa }\left(\sum F^{i}\left(n^{\kappa^{\prime }}\left(\omega_{2}\prime \right)\right)\right),\;\sum_{\kappa^{\prime }=\;1}^{\kappa }\left(\sum \tau^{i}\left(n^{\kappa^{\prime }}\left(\omega_{2}\prime \right)\right)\right),$ the total electrostatic force and torque that is applied to the COM of the microparts due to the simultaneous activation of the $\left[n^{1}\left(\omega_{2}\prime \right)\ldots \;n^{\kappa^{\prime }}\left(\omega_{2}\prime \right)\ldots n^{\kappa }\left(\omega_{2}\prime \right)\right]$ electrodes, where $n^{\kappa^{\prime }}\left(\omega_{2}\prime \right)$ the ${\kappa^{\prime }}^{th\;}$ electrode of the $\omega_{2}\prime {\;}^{th}$ combination. The output of the algorithm is the group of lists $M_{m_{i}}$ (Equation (8)), that is filtered considering the motion that the micropart has to implement. 

### 4.3. Translation of a Rectangular Micropart between Two Equilibrium Positions on a Simulated SP with the Activations that are Camputed by the SEA and MEA Algorithms

In order to certify the efficiency of the SEA and MEA algorithms, a “smart platform” with a grid of 10 × 10 circular conductive electrodes with radius r= 25 μm and $d_{1}=0.7,$ so that $d_{e}=67.5\;\mu m$ and $L_{P}=692.5\;\mu m$ is simulated in Matlab/Simulink. The GCS of the SP is located on the (0,0) μm and a rectangular micropart $m_{1}$ with $L_{m}=6r+3d_{1}r=202.5\;\mu m,$ thickness $\frac{L_{m}}{10}≅\;20\;\mu m$ made of plexiglass with density $\rho_{plexiglass}=1.18\;\cdot 10^{6}\frac{\mu g}{{\mu m}^{3}}$ and weight W=0.08 μN is static on top of the SP and its configuration is given by $q^{1}=\left(-25\;\mu m,\;33.75\;\mu m,\;40{}^{\circ}\right)$ as it is shown in Figure 10a. When the micropart leaves its static condition, the dynamic model for the motion and the rotation of the micropart is given by:(9)$$M_{1}\ddot{r^{1}}+b\cdot \dot{r^{1}}=\sum F^{1},$$
(10)$$I_{1}\ddot{\theta^{1}}+\frac{b}{M_{1}}\cdot I_{1}\cdot \dot{\theta^{1}}=\sum \tau^{1},$$
where $M_{1}$ is the mass of the $m_{1},\;r^{1}=\left(x^{1},\;y^{1}\right),$ the vector that describes the position of the micropart, b the damping coefficient that is computed with respect to the dimensions of the micropart and the viscosity of the $Al_{2}O_{3}$ liquid dielectric layer [33], $I_{1}$ is the moment of inertia of the $m_{1}$ that is given by: $\frac{1}{6}M_{1}l_{s}^{2},$ where $l_{s}\;\mathrm{is}$ the length of the rectangular micropart. The static friction and static torque stop being applied when the micropart leaves its static condition and the damping force $b\cdot \dot{r^{1}}$ and the damping torque $\frac{b}{M_{1}}\cdot I_{1}\cdot \dot{\theta^{1}}$ is presented between the liquid dielectric layer and the component. 

In order to implement the motion of the micropart avoiding the breakdown of the dielectrics [39], the permitted magnitude of the forces and torques that are applied to its COM are specified. Taking into account the analysis in [25], the magnitude of the induced charge of the SP and micropart electrodes corresponds to $10^{-12}Cb\to pCb.$ Moreover, the maximum orders of magnitude of the electric field in order to avoid the dielectric breakdown of the solid $\mathrm{Si}O_{2}$ and the liquid $Al_{2}O_{3}$ is equal to $10^{7}\frac{V}{m}$ and $10^{9}\frac{V}{m},$ respectively [40]. Considering the force that is determined in Section 2, when the total electrostatic force between the couple of the micropart and SP electrode is maximum, the ratio between the planar and vertical electrostatic forces is equal to $\frac{F^{i}{}_{v}\left(d\left(n,\;i_{v}\right)\right)}{\begin{array}{c} \frac{F_{ver}\left(d\left(n,\;i_{v}\right)\right)}{\mu_{s}} \end{array}}≅0.01.$ Thus, the electrostatic force $\sum F^{1}$ that is applied to the micropart must be less than 10μN so that the maximum electric field to be equal to $\frac{\;10\mu N}{1pCb}=\frac{10^{7}N}{Cb}$ = $\frac{10^{7}V}{m}$.

The activations that result from the computations of the SEA algorithm are studied first and the corresponding $S_{m_{1}}$ group of lists shown in Table 4. Τhe capability of the micropart to leave its static condition due to an SP electrode activation implementing a translational or rotational motion is studied. According to the contents of Table 4 it is considered that most of the activations result in the planar motion of the micropart. Moreover, there are many SP electrodes activations that cannot either translate or rotate the micropart. However, just two of them contribute in the translational motion of the micropart and none to its rotation around a fixed axis that passes through its COM. Among the two activation methods for the translational motion of the micropart, the activation of the 45th SP electrode is selected, since is the one that corresponds to the maximum horizontal force $\sum F^{1}$ and $\sum F^{1}<\;10\mu N.$ The activation of the 45th SP electrode is represented in Figure 10b. 

The 45th electrode is charged with 50 V for 10 ms and the motion of the micropart is simulated. The computations for the $r^{1}$ and $\theta^{1},$ with respect to time that are illustrated in Figure 11a,b, validate that the micropart is moved successfully to the left equilibrating on the 45th electrode without changing its orientation. The simulation results certify that the SEA algorithm correctly computed the forces and torques that are applied to the micropart in order to be translated to a new equilibrium position. The motion of the micropart shows the tendency of the electrode underneath the COM of the micropart to be self-assembled with the 45th SP electrode, and finally to be centralized on it. 

As it was discussed in Section 4.2, the MEA algorithm is called when the activations that are included in the resulting group of lists of the SEA algorithm cannot drive the micropart towards a new equilibrium position following a specific direction. According to the simulation results in Figure 11a, the micropart is translated to the left side of the SP. Moreover, as it is shown in Table 4, the activation of the 55th SP electrode where u=1 corresponds to the up-left motion of the micropart. Thus, it is certified that the two activations of the $S_{m_{1}}$ where u=1 contribute in the left/up translation of the micropart. So, in the case that it is requested the micropart to move up or right, the contents of Table 4 cannot prevent a solution and the MEA algorithm is called. In order to minimize the computational time, the search is implemented firstly for κ∈[2 5]. In the case that there is no solution among them, the MEA is called once when κ=6, 7, 8…σ until an appropriate activation is found. The resulting lists of the $M_{m_{1}}$ group are filtered so that u=1
$\sum_{\kappa^{\prime }=\;1}^{\kappa }\left(\sum F^{1x}\left(n^{\kappa^{\prime }}\left(\omega_{2}\prime \right)\right)\right)>0$ and $\left|\sum_{\kappa^{\prime }=\;1}^{\kappa }\left(\sum F^{1}\left(n^{\kappa^{\prime }}\left(\omega_{2}^{\prime }\right)\right)\right)\right|\leq 10\mu N$ and the results are included in Table 4.

As it is shown in Table 5, among the $\sum_{\kappa \;=\;2}^{5}\left(\begin{array}{c} \sigma \\ \kappa \end{array}\right)$ lists, just four satisfy the requested criteria and the activation of the $46^{\mathrm{th}}\;\mathrm{and}\;47^{\mathrm{th}}$ electrodes corresponds to the maximum $\sum_{\kappa^{\prime }=\;1}^{k}\left(\sum F^{1}\left(n^{\kappa^{\prime }}\left(\omega_{2}\prime \right)\right)\right).$ The $46^{\mathrm{th}}\;\mathrm{and}\;47^{\mathrm{th}}$ SP electrodes activation is represented in Figure 12 and the corresponding determinations of the $r^{1}$ and $\theta^{1}$ with respect to time are shown in Figure 13a,b, respectively. 

The simulation results that are illustrated in Figure 13a,b certify that the micropart is displaced to a new equilibrium position implementing a translational motion remaining stable to its start orientation. In Figure 12b, the final configuration of the micropart is presented where the micropart equilibrates since the sum of the forces and torques that are applied to its COM is equal to zero.

To sum up, the simulation computations prove that the SEA and MEA algorithms successfully compute the forces and torques that are applied to the COM of the micropart. Moreover, it is verified that the $m_{1}$ micropart can be centralized due to the activation of a SP electrode close to its COM. Among the big number of lists of the MEA algorithm, a limited number satisfy the criteria for the micropart’s translational motion to a specific direction. The final equilibrium position of a randomly position micropart can be known when it is going to be centralized on a SP electrode. Differently, the micropart is moved until a configuration is reached where the total forces and torques that are applied to its COM is equal to zero, as in the example that is shown in Figure 12b. 

## 5. An Automated Method for the Simultaneous Centralization and Aligning of Multiple Microparts on the “Smart Platform” 

As it was discussed in Section 4, the SEA and MEA algorithms contribute in computing activations methods for the motion of randomly positioned microparts between two equilibrium positions. But, the parallel motion of multiple microparts cannot be achieved in the optimum cycle-time on the SP, while the components keep their random configuration during the whole manipulation process. Specifically, parameters as the distance between two equilibrium positions, the velocity, the relaxation time between two equilibriums etc., are not the same for all the microparts complicating significantly their motion planning [41]. 

On the other hand, when the microparts are centralized and aligned with the GCS, the complexity of the motion planning problem is much smaller [25]. In this section, an automated method for the parallel centralization and aligning of multiple hexagonal and square microparts is introduced. The centralization of the microparts due to the activation of a SP electrode around the COM of the micropart is searched. The simultaneous centralization of multiple microparts avoiding the collision between is implemented specifying rules for the safe parallel motion of neighboring microparts on the SP. Moreover, after their centralization, the microparts are simultaneously aligned, avoiding the collision between them and the static microparts.

### 5.1. The Neighboring Microparts

As it was discussed in Section 1, the microparts are transmitted on the SP with transfer lines and their positioning on the SP is detected by vision sensors. During the transfer on the SP of two adjacent microparts, $m_{i}$ and $m_{j}$ are localized so that the vectors $r^{i},r^{j}$ of their COM to satisfy the constraint that is given by: (11)$$d\left(r^{i},r^{j}\right)\geq 4\sqrt{2}d_{e}\;\;\;\forall i,j\in \left[1,\;k\right],$$
where $d\left(r^{i},r^{j}\right)$ is the Euclidean distance between their centers and k the total number of the microparts on the SP. The $d\left(r^{i},r^{j}\right)$ is computed considering the geometry of the microparts and their neighbor SP electrodes. As it is shown in Figure 14, since the distance $d\left(r^{i},r^{j}\right)$ satisfies the constraint given by Equation (11), the microparts avoid the collision between them; either they are hexagonal or square. Moreover, the specified distances protect the $m_{j}$ micropart from being influenced by the adjacent activations. Specifically, considering the analysis in Section 3, the distance between the activated SP electrodes for the diagonal motion of the $m_{i}$ micropart and the $m_{j}\prime s$ electrodes (electrodes in red ellipse of Figure 14) is bigger than $d_{e}$ in order to apply sufficient forces to the COM of the $m_{j}$ to leave its static condition. 

On the SP, two in parallel moving microparts are considered as neighbors [42] when the distance between their COM is equal or smaller than $1.1\cdot 4\sqrt{2}d_{e},$ where the $0.1\cdot 4\sqrt{2}d_{e}$ a safety distance error. For every micropart $m_{i},$ a group of lists $neighbors^{m_{i}}$ is created where each list is given by:(12)$$\left[m_{j},\;q^{j},d\left(r^{i},r^{j}\right),\;\gamma \right]\;\in neighbors^{m_{i}},\;$$
$$when\;\;d\left(m_{i},m_{j}\right)\leq 4\sqrt{2}d_{e}+0.1\cdot 4\sqrt{2}d_{e},\;\gamma \in \left[1\;n_{m_{i}}\right],$$
where $m_{j}$ is a neighbor of $m_{i},\;q^{j}$ the $m_{j}^{\prime }s$ configuration, γ is an indicator that specifies the number of each list, and $n_{m_{i}}$ the total number of the lists.

### 5.2. The “Free Direction”

During their parallel motion, the microparts have to follow “safe” directions in order to avoid the collision between them. Thus, for each $m_{i},$ four quartiles are specified, where the localization of its neighbors in each of them is defined with the contribution of the angle $\theta^{i,j}.$ The $\theta^{i,j}$ angle given by:(13)$$\theta^{i,j}=tan^{-1}\left(\frac{y^{j}-y^{i}}{x^{j}-x^{i}}\right),$$
where $m_{j}\;\mathrm{is}$ a neighbor of the $m_{i}.$ The $u_{dir}\left(\theta^{i,j}\right),\;dir\in \left\{Up,\;Down,\;Left,\;Right\right\}$ is a function that computes numerically the location of each $m_{i}\prime s$ neighbor in the corresponding quartile. In Table 6, the $u_{dir}\left(\theta^{i,j}\right)$ functions are included and their corresponding graphs are shown in Figure 15. The motion of each micropart is specified by the rules that are included in Table 7, with respect to the $u_{dir}\left(\theta^{ij}\right)$ function. 

An example is shown in Figure 16 for the application of the rules that are included in Table 7. In Figure 16a–c, three examples for the application of the rules included in Table 7 are illustrated. In Figure 16a, the free directions for the $m_{1}$ micropart in the ${\mathrm{UR}}^{1}$ must be found based on the configuration of the $m_{2}$ neighbor. For $\theta^{12}=\;80^{\circ },$
$u_{U}\left(\theta^{1,2}\right)=\frac{80^{\circ }}{90^{\circ }}=0.88>0.5,$ and $u_{R}\left(\theta^{1,2}\right)=-\frac{80^{\circ }}{90^{\circ }}+1=0.12<0.5$ thus, the dir=R is “free” and dir=U, is “occupied”. In the case shown in Figure 16b, the $m_{3}$ neighbor of the $m_{1}$ micropart is positioned next to $m_{2},$ and finally, the $m_{1}$ micropart cannot move in the ${\mathrm{UR}}^{1}$ since $\theta^{1,3}=25^{\circ }$
$u_{U}\left(\theta^{1,3}\right)=\frac{25^{\circ }}{90^{\circ }}=0.28\;\langle 0.5,\;u_{R}\left(\theta^{1,3}\right)=-\frac{25^{\circ }}{90^{\circ }}+1=0.72\rangle 0.5$ (dir=R is “occupied”). In the case that $\theta^{12}=45^{\circ }$ (shown in Figure 16c), both dir=R and dir=U are considered as “occupied” since $u_{U}\left(\theta^{1,2}\right)=\frac{45^{\circ }}{90^{\circ }}=0.5,$
$u_{R}\left(\theta^{1,2}\right)=-\frac{45^{\circ }}{90^{\circ }}+1=0.5,$ and the motion either to R or U cannot certify that the micropart can move in secure. 

For each of the $m_{i}\prime s$ neighbors, the $\theta^{i,j}$ is computed and since there is a dir that $u_{dir}\left(\theta^{ij}\right)\;\geq 0.5,$ the list $\left[Quartiles^{i},dir\;\right]$ becomes a member of the group $FD^{i}$ that includes the occupied quartiles and directions of each micropart. 

### 5.3. A New Method for the Simultaenous Centralization of Multiple Randomly Positioned Microparts on the “Smart Platform”

In this section, a new method for the simultaneous centralization and aligning of multiple randomly positioned microparts is proposed. The rotation of the microparts around a fixed axis that passes through their COM can be implemented since they are centralized [43]. Therefore, the centralization has to be accomplished first applying next the activation algorithms for the alignment of the microparts. According to the analysis in Section 3 and Section 4, a micropart is considered centralized due to SP electrode activation around its COM when the overlapping between the electrode underneath its COM and the activated SP electrode is bigger than 0% and smaller than 95%. The SP electrodes activation methods for the aligning of the centralized microparts are given by Equation (14) considering as n the SP electrode underneath their COM (Figure 17a,b):(14)n, n−N−1, n−N+1, n+N−1,n+N+1.

The centralization/aligning process is divided into manipulation periods MP∈[1 K], where K is the total number of MPs. All the microparts that are not centralized/aligned are included in the $NC^{MP}$ group of lists so that $\left[m_{i}\;p_{1}\right]\;\in NC^{MP},$ where $p_{1}\;\mathrm{is}$ an indicator. The static obstacles space $C^{obs}$ is given by:(15)$$C^{obs}={\left(x-x^{i}\right)}^{2}+{\left(y-y^{i}\right)}^{2}={\left(4d_{e}\right)}^{2}\;\forall m_{i}\notin NC^{MP},$$
where $m_{i}\;\mathrm{is}$ a micropart that is already centralized and aligned and it is enclosed in a disc area with radius equal to $4d_{e}.$ In Figure 18, the collision between the microparts when they reach a configuration in the $C^{obs}$ space is noticed with the red dashed circles. However, in the case that the microparts are centralized on a SP electrode out of the $C^{obs},$ the collision between the microparts is avoided for all the possible orientations of the microparts. 

The group of microparts $NC^{MP}$ is considered firstly and it is called until it is empty. For each micropart $m_{i}$ of the $NC^{MP}$ group, the $S_{m_{i}}$ list is computed with the SEA algorithm. As it is illustrated in Figure 19, the SEA search is limited in a $D_{1}$ disc area around the microparts’ COM with radius equal to 2r, including all the possible electrodes where it can be centralized. Taking into account the analysis in Section 5.2, the microparts cannot be centralized on a SP electrode that corresponds to an occupied direction. Thus, the criterion for the filtering of the $S_{m_{i}}$ group of lists is the collision avoidance between the neighboring microparts. So, for each $S_{m_{i}}\left(\sigma_{1}\right)$ list, the angle $\theta^{\sum F^{i}}\left(\sigma_{1}\right)$ is determined by: (16)$$\theta^{\sum F^{i}}\left(\sigma_{1}\right)=\;tan^{-1}\left(\frac{\sum F^{iy}\left(\sigma_{1}\right)}{\sum F^{ix}\left(\sigma_{1}\right)}\right),\;\;\left|\sum F^{i}\left(\sigma_{1}\right)\right|>F_{s}\;and\;\left|\sum \tau^{i}\left(\sigma_{1}\right)\right|<T_{s.}$$

Then the function $u_{dir}\left(\theta^{\sum F^{i}}\left(\sigma_{1}\right)\right)$ (Table 6) is determined, and considering the rules that are included in Table 7, it is specified if the corresponding direction is occupied or not. The $\left[Quartile^{i},dir\;\right]$ list is created for the corresponding $u_{dir}\left(\theta^{\sum F^{i}}\left(\sigma_{1}\right)\right)$ and since it is not included in the $FD^{i}$ group then $S_{m_{i}}\left(\sigma_{1}\right)\in center^{i},$ where $center^{i}$ is the output of the method. For each micropart, if $center^{i}=\emptyset ,$ the micropart cannot be centralized during the corresponding $NC^{MP}$ since there is no activation to drive it to a free direction. On the other hand, in the case that the $center^{i}$ has more than one list, the list with maximum $\left|\sum F^{i}\right|$ is selected since it corresponds to the SP electrode that is closer to the micropart’s COM. 

### 5.4. Application of the Centralization/Aligning Method 

The simultaneous centralization and aligning of seven microparts on a 20 × 20 electrodes SP with radius r= 25μm and $d_{1}=0.7$ so that $d_{e}=67.5\mu m,$ is studied with the centralization/aligning method. The blue discs around the microparts that are illustrated in Figure 20 have a radius equal to $\frac{4.4\sqrt{2}d_{e}}{2}$ which is computed considering the neighbors distance that is specified by Equation (12). Thus, the blue discs of Figure 20 that tangent or intersect, correspond to neighboring microparts. Moreover, the yellow discs around the COM of the microparts represent the area where the SEA algorithm expands its search for SP electrodes ($D_{1}$ area with 2r radius). The $center^{i}$ for each MP is included in Table 8 and as it is shown in Figure 21 and Figure 22, the $m_{3}$, $m_{4},$
$m_{5},$ and $m_{7}$ are centralized during the 1st MP.

In the case of the $m_{4},$ in $center^{4}$ a list for the 104th electrode activation where $\theta^{\sum F^{4}}=-72.782^{\circ }$ is included since the $m_{1}$ is positioned so that $-45^{\circ }<\theta^{4,1}<0,$ thus $u_{R}\left(\theta^{4,1}\right)>0.5$ and $u_{D}\left(\theta^{4,1}\right)<0.5.$ The second member of the $center^{4}$ corresponds to the 84th electrode activation and is the one that is selected since it corresponds to the maximum value of $\left|\sum F^{4}\right|.$ Similarly, among the two members of the $center^{3}$ and $center^{6},$ the list for the 69th and the 267th electrode activation is chosen considering the $\left|\sum F^{3}\right|$ and $\left|\sum F^{6}\right|$ value, respectively. 

Taking into account the 2nd and 3rd MP, it is substantiated the contribution of the $C^{obs}$ since centralization positions that were forbidden for the $m_{1},\;m_{2,}$ and $m_{6}$ in the 1^st^
MP are permitted in the next two MPs. The activations for the motions that are illustrated in Figure 21 and Figure 22, prove that there is enough distance between the microparts in order to be implemented their centralization and alignment avoiding the collision between them. The activations that are finally applied to the microparts in each MP are included in Table 9. To sum up, it is certified that the proposed algorithm can find solutions for the simultaneous centralization/aligning of multiple microparts avoiding the collision between them, so that the SP is an autonomous system for the automated centralization and aligning of multiple microparts. 

## 6. Discussion, Conclusions, and Future Work

In this paper, a manipulation method for the simultaneous centralization and aligning of multiple plastic/glass-type randomly positioned microparts on a “smart platform” with electrostatic forces are proposed. The FEM analysis computations certify that the centralization of a micropart due to a single SP electrode can be achieved overcoming the drag forces. All the possible activations that can be applied to the electrodes of the SP in order to move between two equilibrium positions are computed by the proposed SEA and MEA algorithms. The simulation results presented validate the effectiveness of the algorithms to determine the forces and torques that are applied to the COM of the microparts after the proposed activations.

This work suggested solutions for the translation of the randomly positioned microparts between two equilibrium positions remaining stable to its start orientation. Moreover, it is shown that the microparts can be centralized due to an activation of a SP electrode around its COM. In addition, the automated method that is introduced contributes significantly to the autonomous parallel centralization and alignment of multiple randomly positioned microparts. 

The simulation determinations certify that the SP is an autonomous device that can be included in the process chain of the microfactories. Thus, the “smart platform” is under construction starting with a 7 × 7 grid in order to be proven the effectiveness of the proposed activation algorithms and to be shown that the motion of a micropart due to them. Moreover, the future work will be concentrated on methods for the batch parallel sorting and assembly of multiple microcomponents on the SP. 

## Figures and Tables

**Figure 1 micromachines-10-00874-f001:**
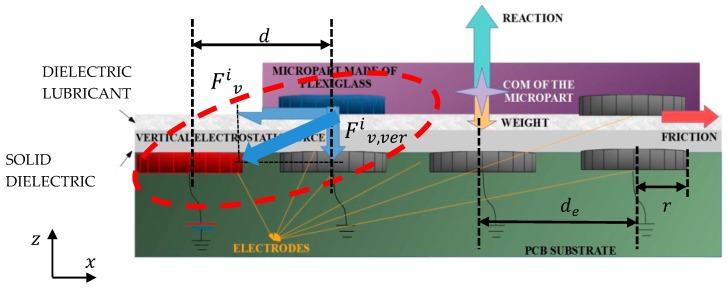
Cross-section of the “smart platform” [25].

**Figure 2 micromachines-10-00874-f002:**
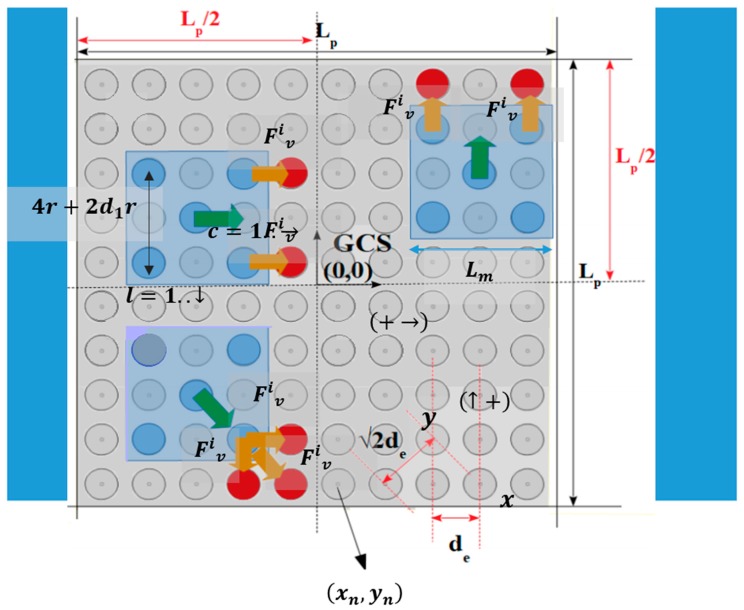
Top view of the “smart platform”. Three centralized and aligned microparts and the corresponding activation methods for the microparts’ horizontal, vertical, and diagonal motion.

**Figure 3 micromachines-10-00874-f003:**
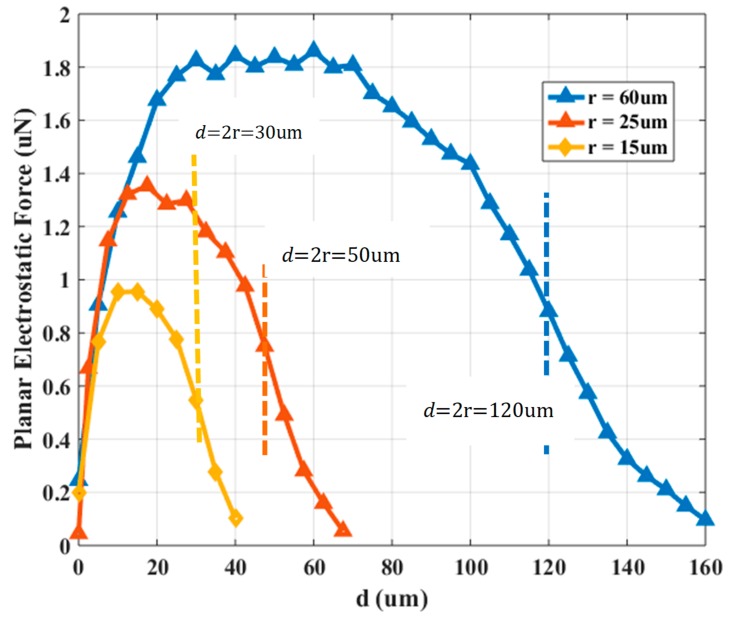
Horizontal electrostatic force ($F^{i}{}_{v}$) between a charged Smart Platform (SP) electrode and an adjacent micropart’s electrode with respect to d.

**Figure 4 micromachines-10-00874-f004:**
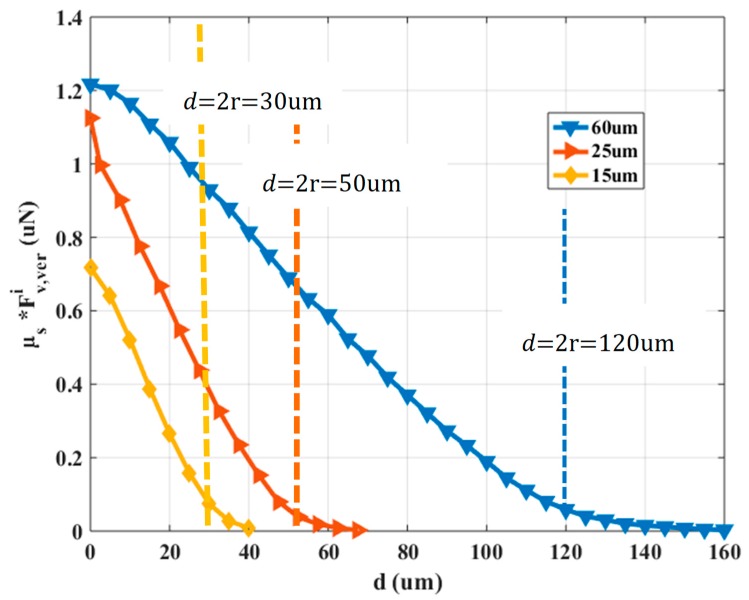
The product $\mu_{s}\cdot F^{i}{}_{v,ver}$ with respect to d.

**Figure 5 micromachines-10-00874-f005:**
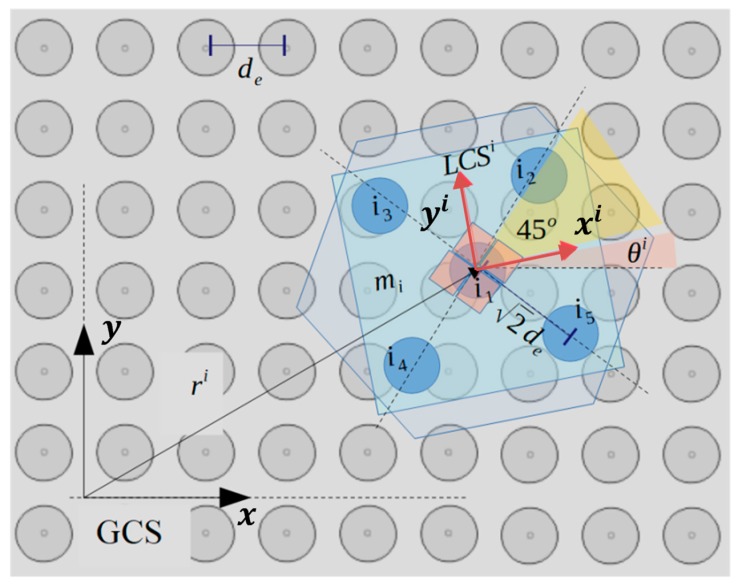
The configuration of a randomly positioned micropart on the SP with respect to the Global Coordinate System (GCS).

**Figure 6 micromachines-10-00874-f006:**
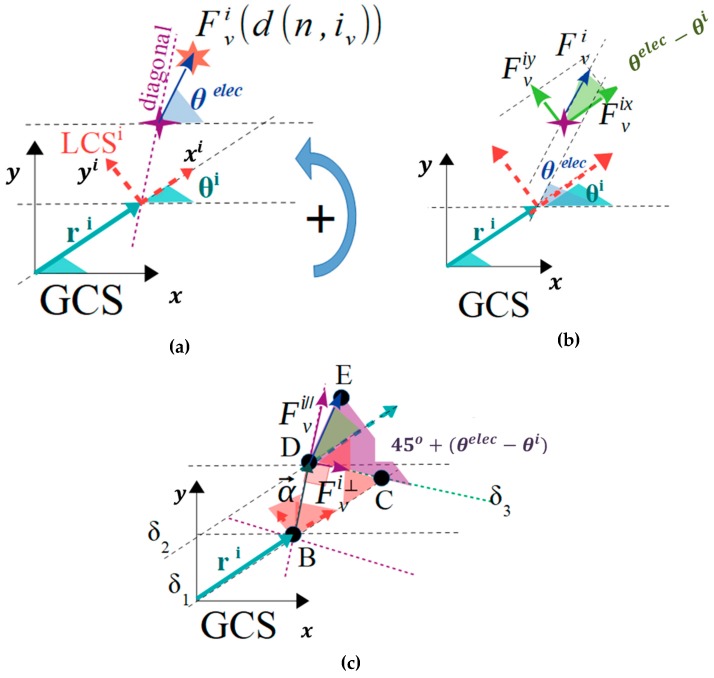
The statics analysis of the micropart (**a**) interaction between the nth activated SP electrode and the i_v_th electrode of the ith randomly positioned micropart, (**b**) contribution of the $F^{i}{}_{v}\left(d\left(n,\;i_{v}\right)\right)$ in the translational motion of the micropart, (**c**) contribution of the $F^{i}{}_{v}\left(d\left(n,\;i_{v}\right)\right)$ in the rotational motion of the micropart.

**Figure 7 micromachines-10-00874-f007:**
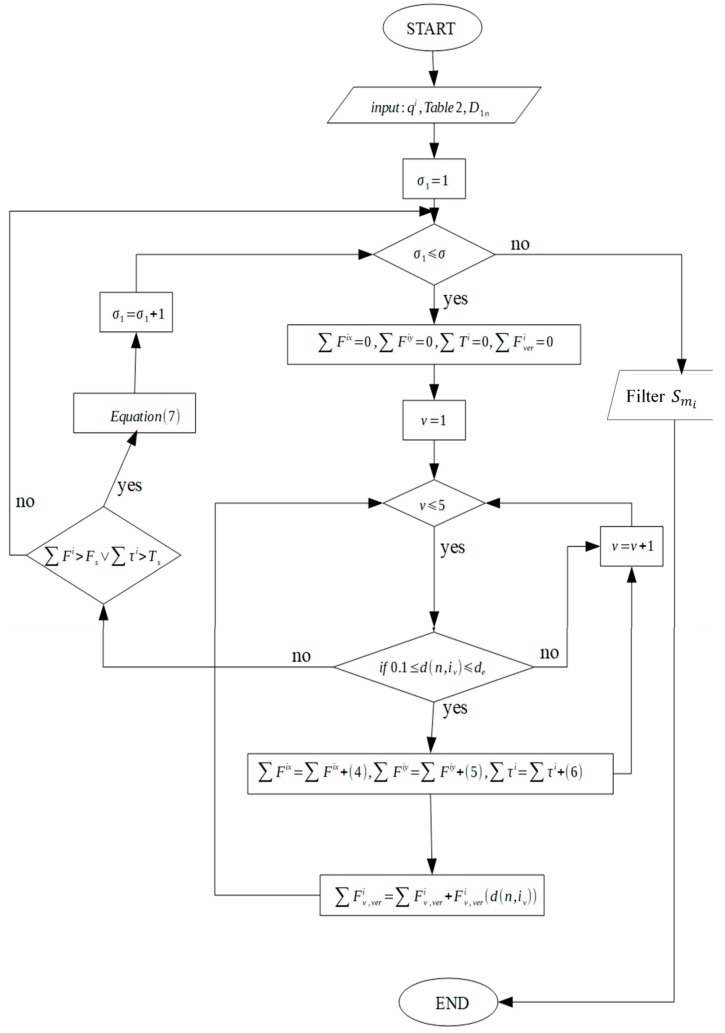
Flow chart of the Single Electrode Activation (SEA) algorithm.

**Figure 8 micromachines-10-00874-f008:**
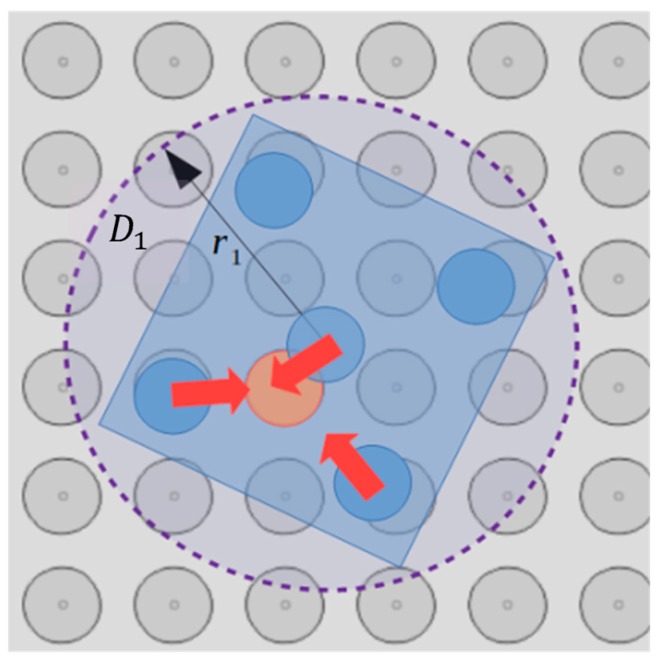
Simultaneous interaction of more than one micropart electrodes due to a SP electrode activation.

**Figure 9 micromachines-10-00874-f009:**
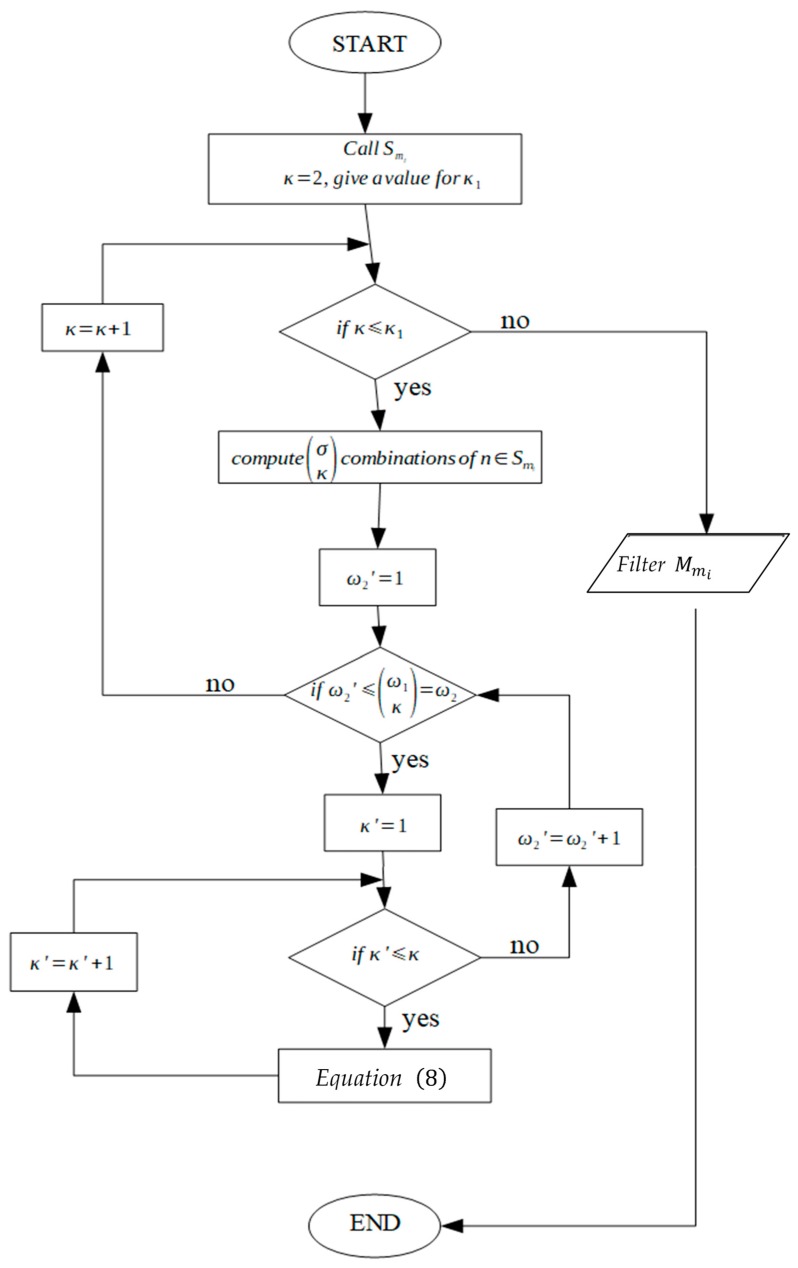
Flow chart of the Multiple Electrodes Activation (MEA) algorithm.

**Figure 10 micromachines-10-00874-f010:**
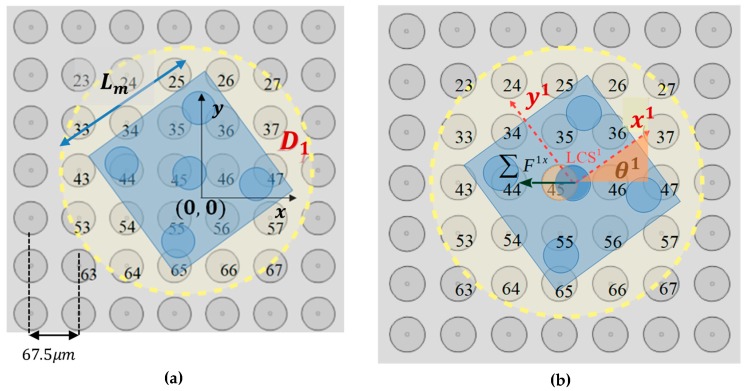
(**a**) Start configuration of the micropart. (**b**) Total electrostatic force that is applied to the COM of the $m_{1}$ due to the activation of the 45th SP electrode.

**Figure 11 micromachines-10-00874-f011:**
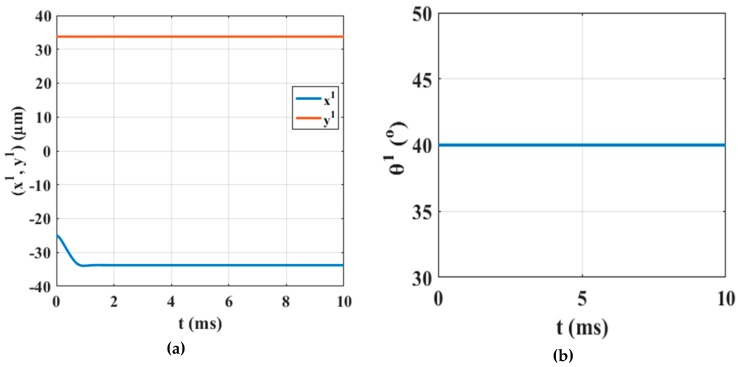
Simulation results for the (**a**) translation and the (**b**) angle of the $m_{1}$ with respect to time due to the activation of the 45th SP electrode.

**Figure 12 micromachines-10-00874-f012:**
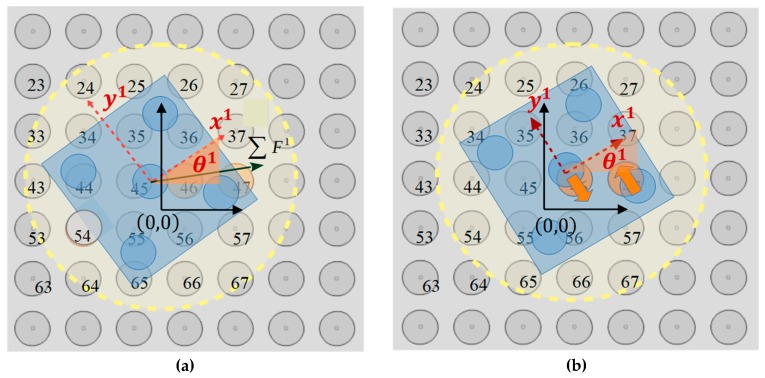
Activation of 46th and 47th SP electrodes (**a**) Activation of the 46th and 47th SP electrode (**b**) Final Equilibrium Position of the $\;m_{1}$ micropart.

**Figure 13 micromachines-10-00874-f013:**
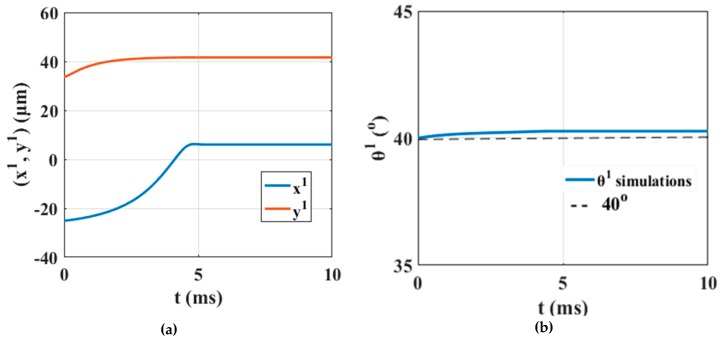
Simulation results of the (**a**) translation and the (**b**) angle of the $\;m_{1}$ with respect to time due to the activation of the 46th and 47th SP electrodes.

**Figure 14 micromachines-10-00874-f014:**
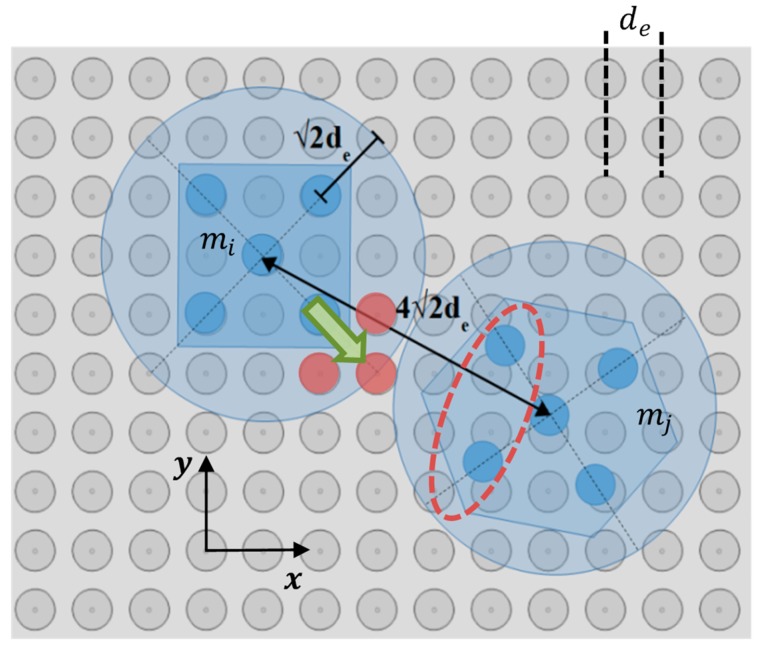
Security distance during the transfer of the hexagonal and square microparts on the SP.

**Figure 15 micromachines-10-00874-f015:**
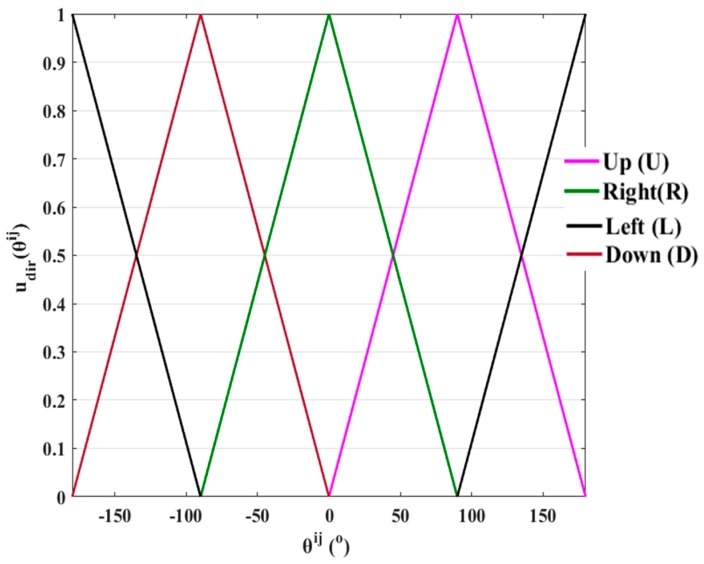
Representation of the $u_{dir}\left(\theta^{i,j}\right)$ function.

**Figure 16 micromachines-10-00874-f016:**
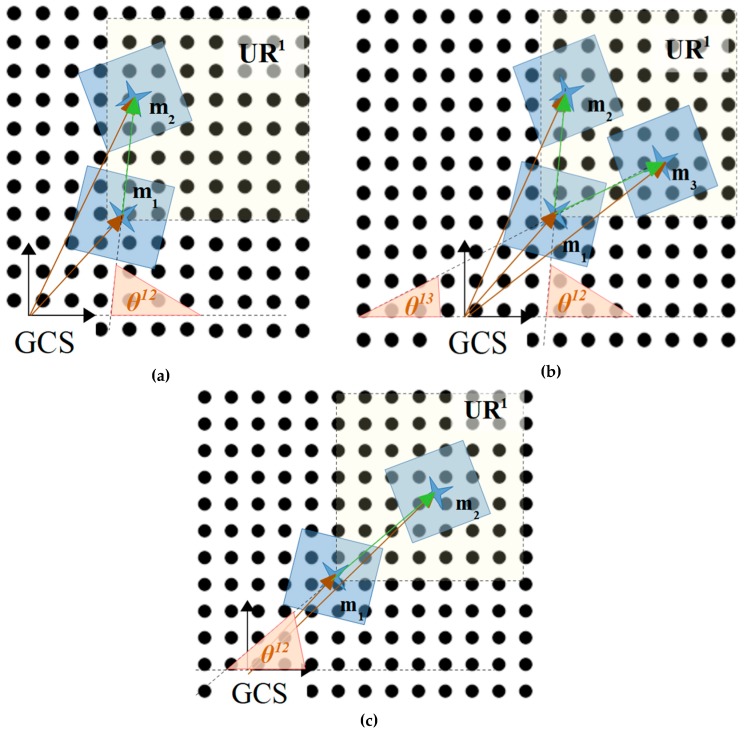
Recognizing the “free directions” in $UR^{1}$ for the $m_{1}$ when (**a**) $\theta^{12}=\;80$°, (**b**) $\theta^{12}=\;80$°, and $\theta^{1,3}=25$°, (**c**) $\theta^{12}=45$°.

**Figure 17 micromachines-10-00874-f017:**
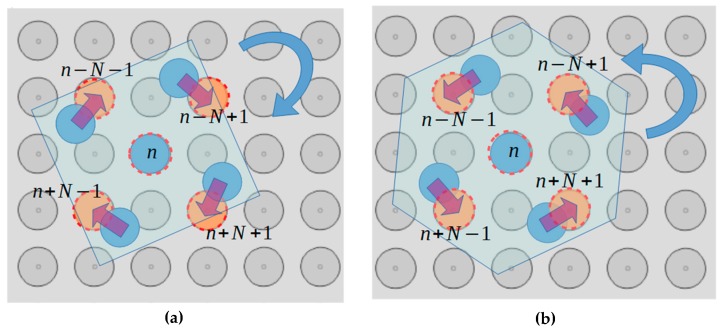
Activation algorithms for the aligning of the micropart on the SP (**a**) Right Rotation (**b**) Left Rotation around the COM of the micropart.

**Figure 18 micromachines-10-00874-f018:**
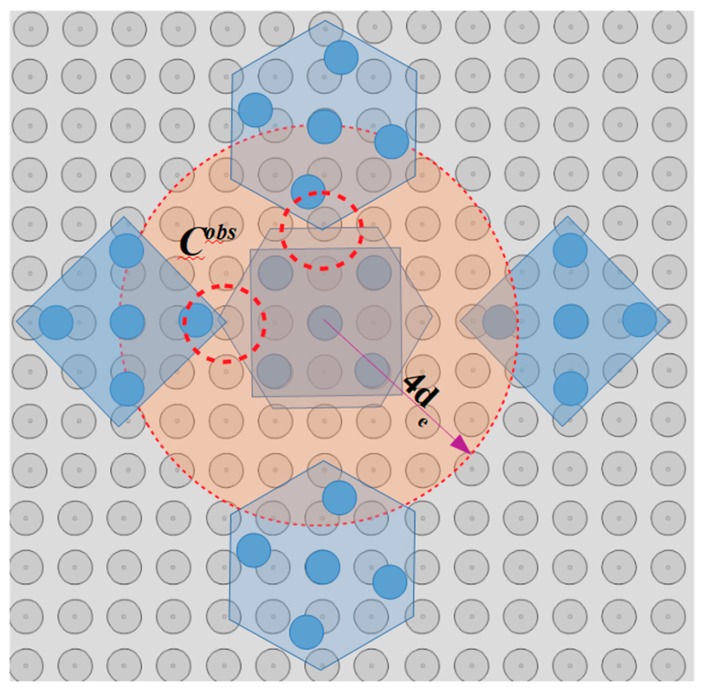
The static obstacles space.

**Figure 19 micromachines-10-00874-f019:**
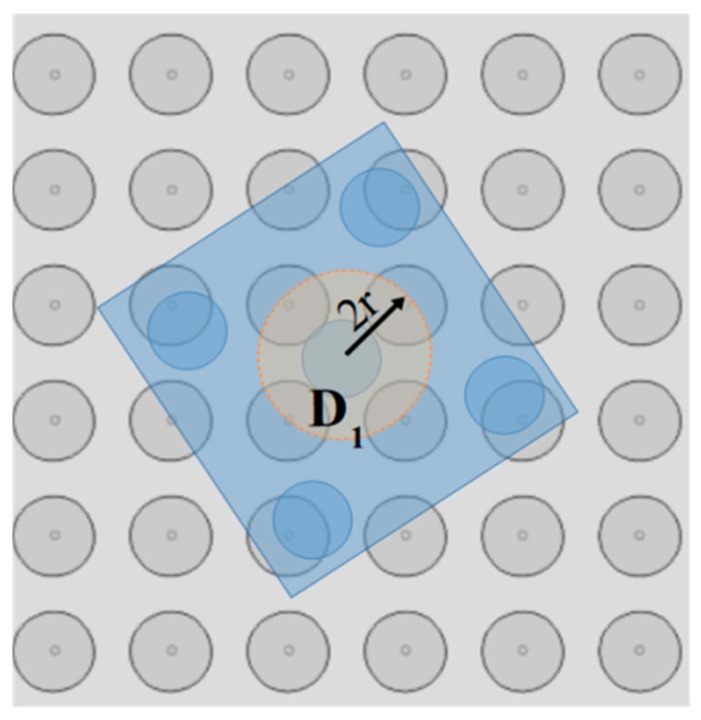
Disc area around the COM of the micropart where the single activation is searched.

**Figure 20 micromachines-10-00874-f020:**
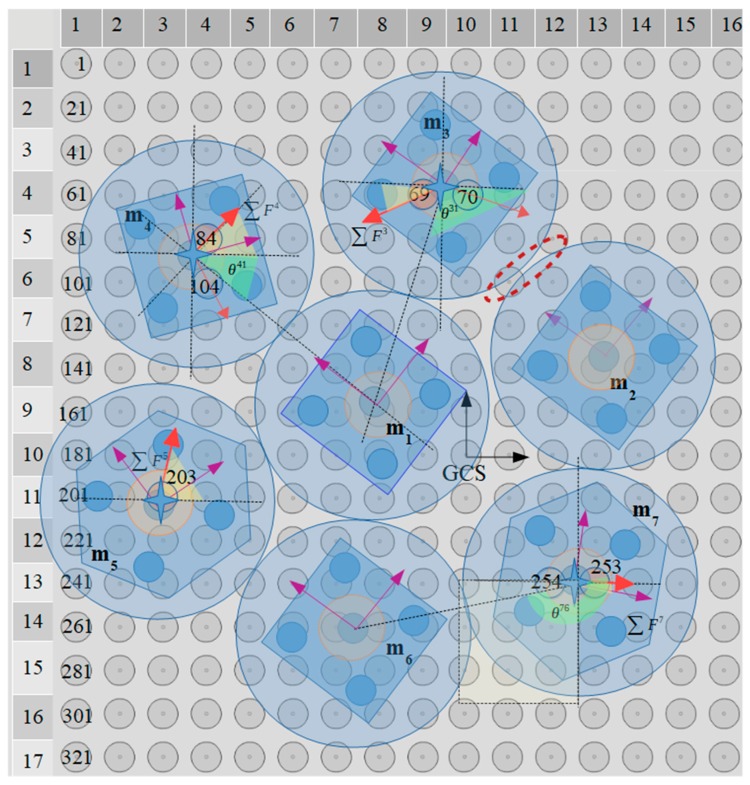
Seven microparts positioned randomly on a 20 × 20 electrodes SP.

**Figure 21 micromachines-10-00874-f021:**
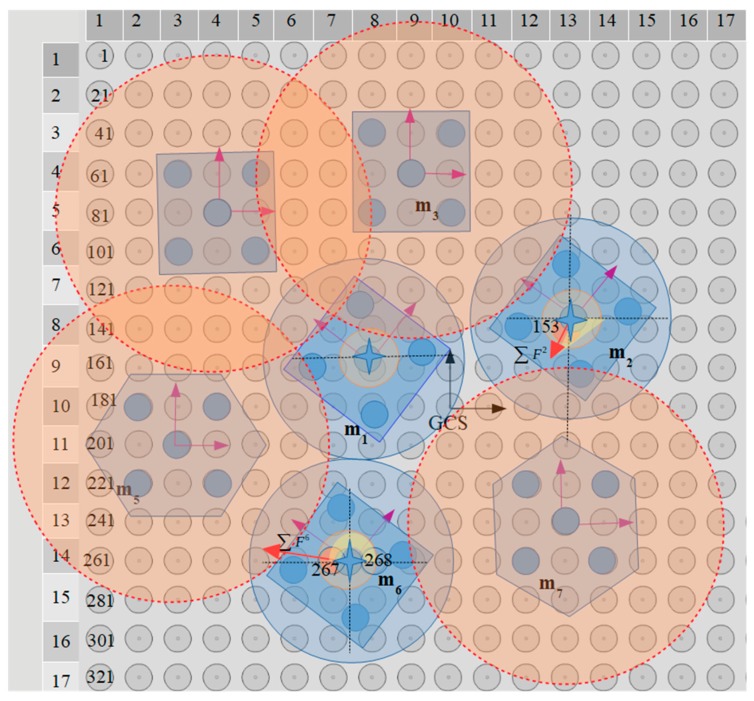
The $m_{3},\;m_{4}$, $m_{5},$ and $m_{7}$ are centralized and aligned and considered as static obstacles when MP=2.

**Figure 22 micromachines-10-00874-f022:**
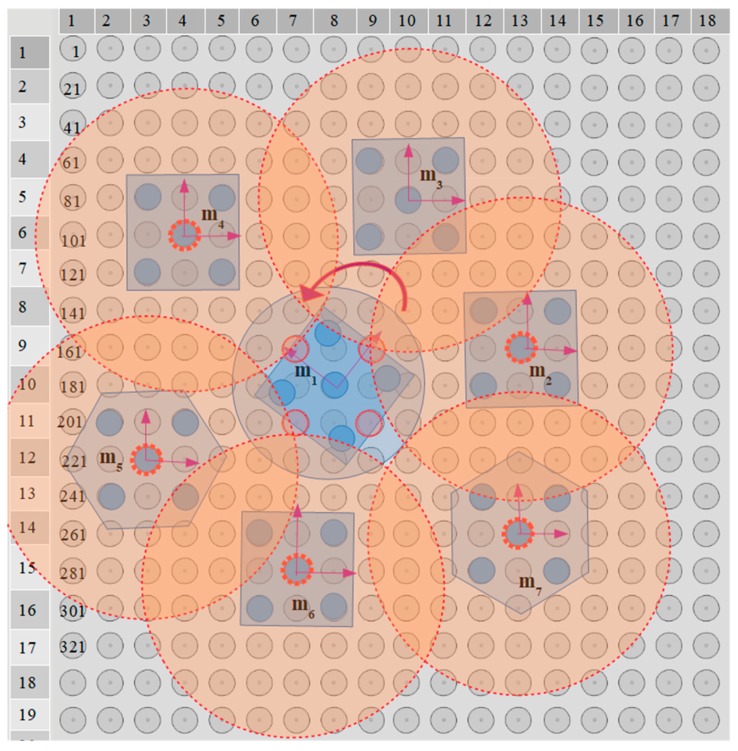
The $m_{1}$ is the last that is centralized and aligned when MP=3.

**Table 1 micromachines-10-00874-t001:** The symbols of the paper and their description.

Symbol	Description
GCS	Global coordinate system
COM	Center of mass
$L_{P}$	Smart platform’s length
n	Number of the SP electrodes
$\left(x_{n\;},\;y_{n\;}\right)$	Coordinates of the center of the $e_{n\;}$
n	Number of SP electrodes
N	NxN grid of SP electrodes
r	Radius of SP and micropart electrodes
$d_{e}$	Euclidean distance between the centers of two neighbor SP electrodes
d	Euclidean distance
	The micropart , i∈[1 k]
k	Total number of microparts that are positioned on the SP
$r^{i}$	Position vector of a micropart on the SP
$q^{i}=\left(\;r^{i},\;\theta^{i}\right)$	Configuration of a micropart on the SP
$LCS^{i}$	Local coordinate system of each micropart
v	Symbol for the number of each electrodes underneath the $m_{i\;}$
$F^{i}{}_{v}\left(d\left(n^{\kappa \prime },\;i_{v}\right)\right)$	The planar electrostatic force between a SP and a micropart electrode
$F^{i}{}_{v,ver}\left(d\left(n^{\kappa \prime },\;i_{v}\right)\right)$	The vertical electrostatic force between a SP and a micropart electrode
$\theta^{elec}$	Direction of $F^{i}{}_{v}\left(d\left(n^{\kappa \prime },\;i_{v}\right)\right)$
$\sum F^{i}$	Sum of forces that are appeared to the center of mass (COM) of the micropart
$\sum \tau^{i}$	Sum of torques that are applied to the micropart
$F_{s}$	Static friction force
$T_{s}$	Static friction torque
SEA algorithm	Single Electrode Activation algorithm
$D_{1}$	Area where is implemented the search of SEA
$D_{1n}$	SP electrodes in $D_{1}$
$S_{m_{i}}$	Group of lists–the output of SEA
$\sigma_{1}$	Number of contents of the $S_{m_{i}}$
MEA algorithm	Multiple Electrodes Activation algorithm
$neighbors^{m_{i}}$	Group of $m_{i}$ neighbors
$u_{dir}\left(\theta^{ij}\right)$	Numerical function for the positioning of the neighbors
$FD^{i}$	Group of forbidden directions
$\theta^{i,j}$	Angle for the positioning of $m_{j\;}$ neighbor with respect to $m_{i\;}$
$M_{m_{i}}$	Group of lists–the output of MEA algorithm
MP	Manipulation period
$C^{obs}$	Static obstacles’ configuration space
$NC^{MP}$	Group of non-centralized microparts
$\theta^{\sum F^{i}}\left(\sigma_{1}\right)$	Angle that specifies the direction of the $\sum F^{i}$
$center^{i}$	Output of the centralization algorithm

**Table 2 micromachines-10-00874-t002:** Equations for the micropart electrodes positioning.

v	$\theta_{i_{v}}$
2	$45^{\circ }+\theta^{i}$
3	$135^{\circ }+\theta^{i}$
4	$225^{\circ }+\theta^{i}$
5	$315^{\circ }+\theta^{i}$

**Table 3 micromachines-10-00874-t003:** Values of the u indicator considering the sum of forces and torques that is applied to the Center Of Mass (COM) of the micropart.

u	Forces and Torques Constraints
0	$\sum F^{i}{}_{v}\left(d\left(n,\;i_{v}\right)+F^{i}{}_{v,ver}\left(d\left(n,\;i_{v}\right)\right)\right)<F_{s}$ and $\sum \tau^{i}<T_{s},$
1	$\sum F^{i}{}_{v}\left(d\left(n,\;i_{v}\right)+F^{i}{}_{v,ver}\left(d\left(n,\;i_{v}\right)\right)\right)>F_{s}$ and $\sum \tau^{i}<T_{s}$
2	$\sum F^{i}{}_{v}\left(d\left(n,\;i_{v}\right)+F^{i}{}_{v,ver}\left(d\left(n,\;i_{v}\right)\right)\right)>F_{s}$ and $\sum \tau^{i}>T_{s}$
3	$\sum F^{i}{}_{v}\left(d\left(n,\;i_{v}\right)+F^{i}{}_{v,ver}\left(d\left(n,\;i_{v}\right)\right)\right)>F_{s}$ and $\sum \tau^{i}>T_{s}$

**Table 4 micromachines-10-00874-t004:** The $S_{m_{1}}$ group of lists for the $m_{1}$ in the $D_{1}$ with $r_{1}\;=2\;\sqrt{2}d_{e}$.

$\sum F^{1x}\;\left(\mu N\right).$	$\sum F^{1y}\left(\mu N\right)$	$\sum \tau^{1}\left(pm\cdot N\right)$	u	n
−0.00226074	0.0047912953	0.1708119	0	24
−0.57322355	2.4521031	−22.41	3	25
0.41062178	0.324912378	5.785385	3	26
−0.059713866	0.04909261	0.7169348	0	33
0.21043153	0.69001	−32.371412	3	34
−1.230287	−0.945225	−25.543675	3	35
1.2234794	0.6597723	28.56835	3	36
0.0125844	0.0306269	0.217871	0	37
−1.740321	−0.63959714	−22.29883	3	43
0.3660568	−1.170881	−91.26852	3	44
−1.489	−0.00146997	−0.86663	1	45
−0.679241	1.13793	42.56993	3	46
0.91002	0.430258	45.2185	3	47
−0.00492973	−0.0076823	−0.18579563	0	53
−0.0341758	0.06120393	5.797169	0	54
−0.399873	0.837	1.87	1	55
0.081896	−0.1542041	−0.12731	0	56
0.1648031	−0.3131	30.2256	3	57
−0.0472974	−0.02778	1.317177	0	64
−0.008132543	−1.5441	30.737	3	65
0.0523	−0.0315323	−5.7058613	0	66

**Table 5 micromachines-10-00874-t005:** Filtered computations of the MEA algorithm considering that u=1,
$\sum_{\kappa^{\prime }=\;1}^{\kappa }\left(\sum F^{1x}\left(n^{\kappa^{\prime }}\left(\omega_{2}\prime \right)\right)\right)>0$ and $\left|\sum_{\kappa^{\prime }=\;1}^{\kappa }\left(\sum F^{1}\left(n^{\kappa^{\prime }}\left(\omega_{2}\prime \right)\right)\right)\right|\leq 10\;\mu N$.

$\sum \left(\sum F^{1x}\left(n^{\kappa^{\prime }}\left(\omega_{2}\prime \right)\right)\right)\;\left(\mu N\right)$	$\sum \left(\sum F^{1x}\left(n^{\kappa^{\prime }}\left(\omega_{2}\prime \right)\right)\right)\left(\mu N\right)$	$\sum \left(\sum \tau^{1}\left(n^{\kappa^{\prime }}\left(\omega_{2}\prime \right)\right)\right)\left(pm\cdot N\right)$	n				
0.4641	0.293	0.0795	26	66			
1.63	−4.6	−0.867	36	37	43	44	45
3.74	1.7	−0.186	46	47			
0.154	−0.649	1.32	56	57	66		

**Table 6 micromachines-10-00874-t006:** The $u_{dir}\left(\theta^{ij}\right)$ functions.

$\mathrm{Angle}\;\theta^{ij}$	$Quartiles^{i}$	$\mathrm{Function}\;u_{dir}\left(\theta^{ij}\right),\;dir\in \left\{Up,\;Down,\;Left,\;Right\right\}$
$0^{\circ }\leq \theta^{ij}\leq 90^{\circ }$	$UR^{i}$	$u_{U}\left(\theta^{ij}\right)=\frac{\theta^{ij}}{90^{\circ }},\;u_{R}\left(\theta^{ij}\right)=-\frac{\theta^{ij}}{90^{\circ }}+1$
$90^{\circ }<\theta^{ij}\leq 180^{\circ }$	$UL^{i}$	$u_{U}\left(\theta^{ij}\right)=-\frac{\theta^{ij}}{90^{\circ }}+2,\;u_{L}\left(\theta^{ij}\right)=\frac{\theta^{ij}}{90^{\circ }}-1$
$-90^{\circ }\leq \theta^{ij}<0^{\circ }$	$DR^{i}$	$u_{D}\left(\theta^{ij}\right)=-\frac{\theta^{ij}}{90^{\circ }},\;u_{R}\left(\theta^{ij}\right)=\frac{\theta^{ij}}{90^{\circ }}+1$
$-180^{\circ }<\theta^{ij}<-90^{\circ }$	$DL^{i}$	$u_{D}\left(\theta^{ij}\right)=\frac{\theta^{ij}}{90^{\circ }}+2,\;u_{L}\left(\theta^{ij}\right)=-\frac{\theta^{ij}}{90^{\circ }}-1$

**Table 7 micromachines-10-00874-t007:** Rules for the safe motion of the microparts in the corresponding quartile.

$Quartiles^{i}$	*Permitted Motions*	*Motion Rules*
$UR^{i}$	Right	$u_{U}\left(\theta^{ij}\right)>0.5,$ $\;u_{R}\left(\theta^{ij}\right)<0.5$
	Up	$u_{U}\left(\theta^{ij}\right)<0.5,$ $\;u_{R}\left(\theta^{ij}\right)>0.5$
	Occupied	$u_{U}\left(\theta^{ij}\right)=\;u_{R}\left(\theta^{ij}\right)=0.5$
$UL^{i}$	Left	$u_{U}\left(\theta^{ij}\right)>0.5,u_{L}\left(\theta^{ij}\right)<0.5$
	UP	$u_{U}\left(\theta^{ij}\right)<0.5,u_{L}\left(\theta^{ij}\right)>0.5$
	Occupied	$u_{U}\left(\theta^{ij}\right)=u_{L}\left(\theta^{ij}\right)=0.5$
$DR^{i}$	Right	$u_{R}\left(\theta^{ij}\right)\langle 0.5\;u_{D}\left(\theta^{ij}\right)\rangle 0.5$
	Down	$u_{R}\left(\theta^{ij}\right)>0.5\;u_{D}\left(\theta^{ij}\right)<0.5$
	Occupied	$u_{R}\left(\theta^{ij}\right)=u_{D}\left(\theta^{ij}\right)=0.5$
$DL^{i}$	Left	$u_{D}\left(\theta^{ij}\right)>0.5,u_{L}\left(\theta^{ij}\right)<0.5$
	Down	$u_{D}\left(\theta^{ij}\right)\langle 0.5,u_{L}\left(\theta^{ij}\right)\rangle 0.5$
	Occupied	$u_{D}\left(\theta^{ij}\right)=u_{L}\left(\theta^{ij}\right)=0.5$

**Table 8 micromachines-10-00874-t008:** Output of the centralization/aligning algorithm for the seven microparts on the 20 × 20 electrodes SP.

MP	Micropart	$center^{i}$				
		$\sum F^{ix}\;\left(\mu N\right)$	$\sum F^{iy}\left(\mu N\right)$	$\sum \tau^{i}\left(pm\cdot N\right)$	$\theta^{\sum F^{i}}\;{(}^{\circ })$	n
1	$m_{3}$	−1.1897	−0.838	0.2235	−159.04	69
		0.81853	−0.581	−0.1235	−26.1	70
	$m_{4}$	0.401	0.82112	0.2253	53. 13	84
		−0.45	0.14	0.1857	−72.782	104
	$m_{5}$	0.9714	1.0491	0.2153	80.88	203
	$m_{7}$	0.754	−0.1396	0.1253	−8.602	253
2	$m_{2}$	−1.33287	−0.2357	−0.1235	−103.05	153
	$m_{6}$	−1.59127	0.695	−0.251	170.1	267
		0.81913	0.1295	0.1325	3.3821	268
3	$m_{1}$	0.5297	−1.0938	−0.2215	60.1	168

**Table 9 micromachines-10-00874-t009:** Activations that are finally applied to the microparts at each MP.

MP	Microparts	n Centralization	n Align
1	$m_{3}$	69	69, 48, 50, 88, 90
	$m_{4}$	84	84, 63, 55, 103, 105
	$m_{5}$	203	203, 182, 184, 222, 224
	$m_{7}$	253	253, 232, 234, 272, 274
2	$m_{2}$	153	153, 132, 134, 172, 174
	$m_{6}$	267	267, 246, 248, 266, 268
3	$m_{1}$	168	168, 147, 149, 187, 189

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
