# Peer review of "Algorithms for the Motion of Randomly Positioned Hexagonal and Square Microparts on a “Smart Platform” with Electrostatic Forces and a New Method for Their Simultaneous Centralization and Alignment"

_micromachines, 2019, doi:10.3390/mi10120874_

Round 1
Reviewer 1 Report
The authors have demonstrated their work on novel algorithms for micro-manipulation. Overall, the reviewer appreciates the significance of the effort since manipulation on the micro-scale is challenging, but with great applications such as MEMS and advanced micro-, nano-manufacturing, also like bio-manipulation. As far as reviewer's knowledge, two major comments on this draft are listed as following:
(1) The first novelty claimed for the algorithm is adaptable for random initial position in workspace. This is not firstly addressed by the authors so far. Then how the strategy handling random parts first is to align and centralize the piece to the electrode, this seems not novel either. The second novelty of the work automated method for "special manipulation" such as parallel manipulation. Again, this is also not firstly addressed by the authors, discussion and/or literature review is required here.
To summarize this comment, authors could do a better job to clearly emphasize the significance and novelty of the algorithm/manipulation strategies.
(2) The author should clearly indicate the targeting dimension of the manipulation. For example, when objects go to micro-scale, the inertia/mass is not the dominating factor anymore, where the force, XXXXX Then how well the algorithm and simulation will predict the real performance could be questionable, especially when missing preliminary physical testing, even though we don't have a prototype so far with the full grid design. Lacking work on this aspect will further impact the potential if authors would prefer to conduct the manipulation on smaller micro-scale.
Other minor comments are:
(1) In Figure 1, a photo of real device would be good. Even though we don't have physical device so far, a top view of schematic would also be more intuitive, like as Figure 1 (b).
(2) Tables, flowcharts, fonts should be carefully edited, at least uniform.
(3) Although the dimensions of the grids and electrodes are mentioned in the text, the annotation of dimension on the schematic figure would still be good, such as figure 2, 3, 5, etc.
(4) Some wording is relatively casual for a technical paper. For example, in section 2, line 124 and 125, "special manipulation". Also like in section 5, line 358, "not that possible".
(5) Missing reference 1, 2.
Author Response
Dear reviewer,
Thank you for your comments. "Please see the attachment."
We are looking forward to your reply.
Kind regards,
The authors

Reviewer 2 Report
The manuscript presents manipulation methods of small parts by using Smart Platform. Authors performed some analysis to determine the orbital of the parts. However, target and novelty of the work are not clear. Please confirm following comments.
Target of this method is not clear. Authors mentioned general background of the MEMS assembly in introduction. However, the proposed method uses microparts with five electrodes and substrates with electrode array covered with dielectric materials. What is the target for such a specialized system? Does proposed algorism have generality applicable for parts with any position? Can authors proof this? How do you determine the friction? If the friction value changes, the proposed method is available with same mechanical model? Practically, how do you detect the position and angle of parts? I believe this information is needed for this method.
Followings are minor points.
Define COM in abstract. There are many grammatical error and readability of the English is poor. Reference start from “3”.Author Response
Dear reviewer,
Thank you for your comments. "Please see the attachment."
We are looking forward to your reply.
Kind regards,
The authors

Round 2
Reviewer 2 Report
Authors did not properly answer my question 1, 2, and 4.
I thinks the manuscript does not have enough quality for publication.